# Telomere Dynamics in Livestock

**DOI:** 10.3390/biology12111389

**Published:** 2023-10-31

**Authors:** Nan Zhang, Emilie C. Baker, Thomas H. Welsh, David G. Riley

**Affiliations:** 1Department of Animal Science, Texas A&M University, College Station, TX 77843, USA; nanzhang0828@tamu.edu (N.Z.); thomas.welsh@ag.tamu.edu (T.H.W.J.); 2Department of Agricultural Sciences, West Texas A&M University, Canyon, TX 79016, USA; ebaker@wtamu.edu; 3Texas A&M AgriLife Research, College Station, TX 77843, USA

**Keywords:** telomere, longevity, livestock, cattle, horses, pigs, sheep, goats

## Abstract

**Simple Summary:**

Short repeated sequences of DNA at the end of chromosomes are called telomeres; these help to maintain the chromosome structure. During normal cell replication, some of the sequences are lost; therefore, the length of telomeres may be associated with the aging of the overall animal. Longevity in livestock is difficult to measure and improve; however, telomere dynamics may represent an opportunity to monitor or influence longevity. Beginning in the 1990s, livestock clones were studied with respect to the change in telomeres because cloned DNA was often from a mature animal and, therefore, might have shorter telomeres associated with age. However, results differed by the species and cell type used to produce the clones. Telomere dynamics have been best characterized in dairy cows, pigs, and horses. In general, older animals have shorter telomeres, and stress imposed upon animals results in shorter telomeres; however, exceptions have been observed to both generalities. Attrition and other telomeric characteristics have been less well described in beef cattle, goats, and sheep. Longevity is measured uniquely and differs for species and production systems within the same species; therefore, it is appropriate to encourage the telomere-longevity correspondence in each.

**Abstract:**

Telomeres are repeated sequences of nucleotides at the end of chromosomes. They deteriorate across mitotic divisions of a cell. In *Homo sapiens* this process of lifetime reduction has been shown to correspond with aspects of organismal aging and exposure to stress or other insults. The early impetus to characterize telomere dynamics in livestock related to the concern that aged donor DNA would result in earlier cell senescence and overall aging in cloned animals. Telomere length investigations in dairy cows included breed effects, estimates of additive genetic control (heritability 0.12 to 0.46), and effects of external stressors on telomere degradation across animal life. Evaluation of telomeres with respect to aging has also been conducted in pigs and horses, and there are fewer reports of telomere biology in beef cattle, sheep, and goats. There were minimal associations of telomere length with animal productivity measures. Most, but not all, work in livestock has documented an inverse relationship between peripheral blood cell telomere length and age; that is, a longer telomere length was associated with younger age. Because livestock longevity affects productivity and profitability, the role of tissue-specific telomere attrition in aging may present alternative improvement strategies for genetic improvement while also providing translational biomedical knowledge.

## 1. Introduction

Understanding telomere dynamics, and especially its relationship with longevity or survival traits, may enhance livestock improvement. Genomic strategies for the improvement of livestock have developed substantially since the original sequencing of reference genomes in multiple species. As these mature, they are mostly based on the prediction of genetic merit, but functional annotation of genomes is a worthy effort. Both could and/or do serve for the identification of animals to selectively improve traits of economic importance in the production of protein and fiber to support humankind. Appropriately, genome organization, arrangement, and structure are being investigated for their potential roles in livestock improvement. Telomeres in mammals are repeats of thymine (*T*), adenine (*A*), and guanine (*G*) bases in the DNA sequence *TTAGG* that occur at the end of chromosomes. These sequences are essential for maintaining chromosome organization and structure in cells. Because of the nature of replication, telomere sequences degrade across mitotic time. In *Homo sapiens* and other species, this degradation has been confirmed (an annual average of 30 to 50 base pairs (bp)) to be associated with cell senescence and mitosis and has been postulated to be correspondent to length-of-life traits. Longevity in livestock is lowly heritable (<0.1); that is, a low proportion of the differences among individuals are due to additive genetic action. Therefore, longevity is slower to respond to selective improvement than traits with higher heritability. Longevity is economically important in livestock production scenarios because time is needed for dams to produce enough progeny to make a return on investment. The objective of this review is to summarize the current knowledge base of telomere dynamics in livestock.

## 2. Telomere Biology

Telomere structure and function have prominent roles in inheritance [1,2,3,4]. Telomeres are conserved DNA structures in animals that consist of multiple copies of a short nucleotide sequence (*TTAGG*) repeated from 100 to 3000 times at the end of chromosomes [5,6,7]. Telomeres are essential components in the inheritance process because they help maintain chromosomal structure and the entire genome within the cell nucleus [8,9]. They have single-strand loop structures with high guanine content that overhang their terminus and consist of a complex structure maintained by multiple proteins [8]. The range of TL is about 3 to 18 kb for *H. sapiens* [10] and ranges from 6 to 30 kb for *Bos taurus* [11] and 7 to 21 kb for *Equus caballus* [12].

In normal mitotic replication and the absence of some counter mechanism, one of the two strands (lagging) of DNA has telomeric loss. This occurs because replication is unidirectional. The leading strand is replicated accurately in the 5′ to 3′ orientation. The lagging strand is fragmented (100 to 200 base pairs in length, called Okazaki fragments), and the 5′ to 3′ replications are subsequently ligated. The single RNA primer on the lagging strand leaves gaps at those ligation points; at the 3′ end of the lagging strand, they cannot be attached, and that gap is not filled, resulting in a reduction of 50 to 250 base pairs. This progressive shortening of telomeres across mitoses is called the “end replication problem”. The “Hayflick limit” represents a point at which cells move to senescence [13,14]. Olovnikov [15,16] postulated a numerical boundary and counting mechanism for potential replications before cell senescence. Hayflick [17] confirmed that human (non-cancer) cells cease division after 40 to 60 times.

This successive loss of DNA in mitotic divisions is somewhat offset by telomerases [18,19,20]. These enzymes offset the telomeric loss of DNA by de novo synthetization of sequences. Another corrective process apparently occurs only in cancer cells and is known as the “alternative lengthening telomeres pathway” [21]. Alternative lengthening of telomeres (ALT) involves the interplay of various proteins facilitating homologous recombination between subtelomeric regions of neighboring chromosomes. While the required proteins for ALT are generally present in most cells, the precise mechanism that inhibits it from maintaining telomere length in all cells is unknown [21]. Telomere length (TL) of leukocyte DNA has been proposed as an indicator for attributes of premature aging [22,23,24] and for mortality [4,25,26]. The “telomere brink” was described by Steenstrup et al. [27] as approximately 5 kb in humans and represented a high risk of death. Louzon et al. [1] detailed published work relative to the effects of environmental influence and stress on TL; often the effect is detrimental: telomeres are shortened [28,29]. However, this is not universal for all studied species and environmental insults; some resulted in a lengthening of the telomeric sequence. Some stress or environmental factors influence the protein components of the telomeric structure as well as the offsetting damage correction mechanisms, rather than a direct influence on the telomere sequence itself [28]. The effects of maintaining telomeres may even begin prenatally [30]. Scientific clarification of the connection between TL variability and biologic consequences proceeds today.

The TL of neonate humans may be associated with parental TL, though it is not conclusive whether it corresponds more closely with paternal or maternal telomere phenotypes [31,32,33]. Lengthening of telomeres in the DNA of spermatozoa may be influential on telomere attrition in the DNA of somatic cells in their offspring; that is, children of older fathers inherit longer telomeres, and this may persist across generations [34,35]. However, these results may be influenced by the birth cohort; there may be confounding of the sperm aging increase in TL with the inheritance of longer telomeres of DNA from younger ovum (female gametes) [36,37]. Environmental and social stressors, such as feed restriction or hierarchy establishment, often elicit increased secretion of adrenal glucocorticoids, which is inversely associated with TL. Mechanistically, it has been suggested that stress-induced glucocorticoids and oxidative stress negatively affect telomere maintenance and mitochondrial function [38]. In view of mitochondria’s major role in energy balance, TL may be subject to chronic exposures to stressors that alter energy balance, particularly at key developmental points during prenatal or postnatal growth phases [39,40,41]. Although TL in leukocytes is classically linked with the lifespan or longevity of many species, it is important to denote that the rate of attrition is greater earlier in life and proceeds in a non-linear fashion in humans and cattle [42,43,44]. Moreover, telomeres also shorten in various tissues with age in humans [45]. By comparison, telomere dynamics have been assessed in far fewer specific tissues in livestock to date as the peripheral leukocytes predominate as the surrogate for internal tissues related to metabolism, immunity, fertility, etc. An individual’s genetics and exposome are two primary determinants of TL. Comprehension of how telomere dynamics are intertwined with health, well-being, performance, and longevity of livestock species depends on the inclusion of tissue-specific analyses. Livestock species models can also serve as a platform to access fetal tissues to identify the mechanisms whereby environmental stressors imposed during gestation induce advantageous or disadvantageous epigenetic alterations upon future generations [46,47,48,49,50].

## 3. Methods to Determine TL

The Telomere Research Network (TRN), sponsored by the National Institute of Aging and the National Institute of Environmental Health Sciences, is “dedicated to facilitating the collaboration between basic telomere biologists, population and exposure researchers, and other scientists across disciplines to advance interdisciplinary research on telomeres as sentinels of environment exposure, psychosocial stress, and disease susceptibility” [51]. The TRN and its members have published position papers that summarize contemporary and emergent methods to monitor telomere dynamics principally in humans and animal model species [52,53,54]. The most recent publications related to telomere dynamics in livestock primarily apply a simplex or multiplex quantitative polymerase chain reaction (qPCR) method to assess TL in peripheral blood monocytes or some tissues. The qPCR method is useful for smaller quantities of DNA, and higher throughput of many samples compared to the Southern blot method. The Southern blot method is often used to assess TL in tissue samples from cloned livestock (Table 1). The number of study subjects, the quantity of DNA available, and laboratory capabilities are factors most often considered in the selection of which calibration method to apply. The published and emerging methods have advantages regarding repeatability, throughput, cost, and usefulness to compare within or across studies or laboratories. Wang et al. [55] compared qPCR and fluorescence in situ hybridization (flow FISH) methods for the estimation of TL using DNA extracted from peripheral blood mononuclear cells of nearly 200 healthy humans who were donors to a bone marrow repository. A modest correlation (r = 0.56) between methods was calculated regarding estimated TL. They offered two major caveats: (1) linear equations used to predict TL from qPCR T/S ratio data sets could lead to a biased result, especially for individual study subjects or patients bearing the shortest or longest telomeres; and (2) use of qPCR T/S ratios to compare results across studies has risk due to potential intra-laboratory variability. As described in greater detail in the following section, Southern blot and qPCR [56] are two of the most popular methods adopted in earlier research. Southern blot analysis employs terminal restriction fragment assays [57], while qPCR analysis utilizes repeated measures of the telomere-to-single-copy-gene ratio [58]. Elbers et al. [59] compared these two methods using data from Kimura et al. [58] and Cawthon [57]. They confirmed that both methods provided acceptable results for showing the effects of aging on TL. Additionally, they found that females usually possess longer telomeres compared to males. Elbers et al. [59] also pointed out a potential disadvantage of using qPCR: it has a larger measurement error compared to Southern blots, with error rates of 5.8% and 1.5%, respectively. Seeker et al. [60] favorably evaluated a calibrated (by the DNA extraction method) qPCR and concluded that membrane-based DNA extraction methods result in shorter TL measurements than others (not based on membrane extraction protocol).

However, the Southern blot assay has its own limitations as well; it requires a certain amount of DNA material, which can be hard to obtain when measuring TL with a small number of cells. Ji et al. [71] tried to adopt a method that is commonly used for checking telomeres in mouse and human cells—telomere quantitative fluorescence in situ hybridization (Q-FISH)—for analyzing porcine telomeres. The assay Q-FISH can provide more specific results at the individual cell level, which is ideal for porcine telomere studies due to its natural instability [72]. Similarly, Russo et al. [73] used Q-FISH to process pig ovarian follicle samples to detect telomerase catalytic subunit activities, which have been shown to be the fundamental mechanism of maintaining TL [74]. They were able to find evidence that telomeres stably get progressively larger from preantral to antral follicles, and specifically pointed out that employing the Q-FISH technique helps them counteract the difficulty of securing sufficient amounts of DNA from oocytes, which is a common problem for Southern blot assays, as previously mentioned. Nevertheless, depending upon the cell type, Southern blot might be a superior choice over assays like Q-FISH because the latter requires an established cell culture, which could be confounding since culturing cells leads to the shortening of TL [64].

Seeker et al. [75] noted the possibility of inaccuracies in telomeric assays due to the presence of interstitial telomeric sequences. The interstitial regions of DNA are telomeric but located somewhere else in the genome [76], and the probe used in assays may accidentally bind to these interstitial telomeric sequences, resulting in errors for the TL measurements [77]. To avoid inadvertently measuring interstitial telomeric sequences, in-gel hybridization techniques can be used in developing terminal restriction fragment assays. In-gel hybridization specifically allows binding to terminal telomeres, excluding interstitial telomeric sequences [78].

The methods discussed thus far are effective for obtaining average or relative TL, but the shortest telomeres can also provide valuable information regarding DNA damage responses, which are often followed by replicative senescence [79,80]. Studies focusing on the measurement of the shortest telomeres may benefit from using telomere shortest length assay (TeSLA) [77]. This assay requires only small amounts of DNA and can help determine longitudinal changes over various time intervals. Compared to data generated by Q-FISH, TeSLA is more effective at distinguishing terminal telomeric repeats from interstitial telomeric sequences. However, TeSLA is not ideal for measuring very long telomeres due to its low throughput [77]. Emergent methods to assess TL in human tissue samples include peptide nucleic acid (PNA) hybridization and analysis of single telomeres (PHAST) [81]. The newly reported methods remain to be validated for application to TL analysis of tissue samples of livestock. An advantage will be the possibility to determine with extreme resolution the length of telomeres of individual specific chromosomes using fluorescence spectroscopy (as few as 100 bp [81]), single-molecule real-time (SMRT) sequencing-based methods (at nucleotide resolution; [82]), or nanopore sequencing (as few as 75 bp at specific ends of chromosomes; [83]). Moreover, efforts are underway to quantitatively determine the TL and telomerase activity by harnessing the CRISPR-Cas (clustered regularly interspaced short palindromic repeats-CRISPR associated proteins) system [84]. Regarding telomere dynamics in livestock, the goals and experimental design of the project should be the overriding factors in deciding which of the current and emergent methods to apply.

## 4. Cloned Animals

The use of somatic cell nuclear transfer (SCNT) in the livestock industry has provided the opportunity to clone genetically superior animals and produce cloned transgenic animals [85]. One concern regarding SCNT is when using an aged somatic cell for cloning the offspring might start life with shortened telomeres, resulting in earlier cell senescence. This premature aging can result in diseases such as cancer being more abundant earlier in life for cloned animals. relative to their non-cloned counterparts. Conflicting results have been reported by researchers who investigated TL in cloned livestock/animals. Dolly was the cloned ewe with DNA from a mammary gland cell of a Finn Dorset ewe [86]. This first successfully cloned animal had significantly shorter telomeres relative to sheep of the same age [67]. It was concluded that Dolly’s telomeres were more consistent with the telomeres of the donor, and the telomeres were not able to restore any length because the cloned animals are produced without germline involvement. TL in cloned sheep has almost always been reported to be shorter than controls [87]. Shorter telomeres relative to age-matched controls were also observed in goat clones derived from adult granulosa donor cells [69,70,88]. In contrast, the telomeres of cloned beef cattle and pigs were the same or even longer than telomeres of the non-clone age-matched animals [61,64,89].

One potential reason for these conflicting results is that the cell type used for the SCNT could influence the TL of cloned animals. Bovine clones derived from oviductal and mammary epithelial cells from a 13-year-old Holstein cow had significantly shorter telomeres relative to age-matched controls and to 18-year-old control animals [63]. Similar results were observed when using oviductal epithelial cells from 6-year-old Jersey cattle. However, when muscle cells from a 12-year-old bull were used, the TL of cloned bulls were not significantly different relative to age-matched controls [63]. Embryonic cell-cloned cattle (using nuclei of 28- to 49-cell stage blastomeres) and clones produced from fibroblasts exhibited significantly longer telomeres than age-matched controls [62]. In sheep derived from two different fibroblast cultures, one early passage culture (minimal transfers to new medium) and one late passage (numerous transfers to new medium) the clones from the earlier passage culture TL were restored. In some incidences, the telomeres of the clones were longer than those of the donor cells. Clones derived from the late passage fibroblast exhibited telomere lengthening but not complete restoration [68]. Despite the research in other livestock species, there has not been any documented description of TL dynamics in cloned horses [90].

Due to the observed variability of TL, Tian et al. [61] hypothesized that the extent of reprogramming (lengthening) of telomeres depends upon the extent of telomerase activity during the transfer and culture process. As the nuclear donor cells do not undergo gametogenesis, the telomere reprogramming/lengthening after SCNT must occur during embryo development. To investigate this, Tian and colleagues investigated telomerase activity at different stages of in vitro cloned bovine embryo development. A gradual increase in telomerase activity was observed from the oocyte to the morula stages of development, with a large increase in activity at the blastocyst stage [61]. These results were validated by the reprogramming of telomerase activity and subsequent telomere lengthening in the blastocyst stage in cloned cattle, where again, no difference in TL difference was observed between the clones and the age-matched controls [88]. Tissue-to-tissue variation in TL was also observed in the clones.

Variations in telomerase activity between cells could be responsible for the differences in TLs discussed previously. Telomerase activity varied between tissue types in humans and mice [91]. Variability in TL in clones could also be due to species and sex differences. In pigs, cloned individuals exhibited longer telomeres relative to age-matched controls, while in cloned cattle, there was no difference in TL [65]. Increased telomerase activity was also observed in pig SCNT blastocysts compared to donor cells and blastocysts derived from in vitro fertilization, but the same pattern was not observed in bovine SCNT blastocysts. In goats, female clones exhibited shorter telomeres than the donor, while there were no differences in length between male clones and their donors [70]. The protocol used in SCNT might also influence TL. Trichostatin A (TSA), a histone deacetylase inhibitor, has improved the development and viability of cloned embryos. Treatment of porcine embryos with TSA resulted in longer telomeres in the cloned offspring relative to the donor cells. Embryos treated with TSA also had increased mRNA levels of telomerase relative to non-treated cells [66]. Additionally, the small sample size and large inter-individual variation in TL of cloned individuals have made accurate characterization of the effects of SCNT on TL difficult. 

In summary, to date, numerous factors, from donor cell type to SCNT protocol, influence TLs have been identified in cloned livestock species. Continued research utilizing larger sample sizes and numerous cell types and telomerase activity analysis is needed to fully understand how cloning affects TL and, therefore, livestock health, performance, and longevity.

## 5. Telomere Research in Livestock

Table 2 summarizes the major telomere characterization efforts by species in livestock. Contents are drawn from 12 studies about dairy cattle, 2 about beef cattle, 2 about pigs, 3 about sheep, and 6 about horses. Their findings will be discussed in detail in the corresponding sections. 

### 5.1. Dairy Cattle

Brown et al. [92] reported results from the assessment of TL in Holstein cows (n = 201 from 10 herds; herd effect was important). Cows were sampled a single time. The average TL was lower for cows at older ages and for cows that left the herd early (within one year of sampling). 

TL in Montbéliarde and Holstein cows, and F_1_ crosses of these breeds were examined (n = 747 cows in two herds). Although a goal of interest was to assess the relationship of heterosis with TL and other traits, no breed differences were detected [93]. Breeding values for TL from whole blood measured before calving and during lactation were associated with improved production (“livability” and “productive life”) and higher resistance to retained placenta, mastitis, metritis, and displaced abomasum, but not cow longevity nor other measures of fitness [94]. Estimates of heritability ranged from 0.12 to 0.2. Although there was high individual variability of TL, in general, TL declined with age [93,94]. 

Telomeres have been characterized in the DNA from leukocytes of Holstein–Friesian cows in Scotland [75,95,96,97,98]. Annual blood samples for assays and records were obtained for 308 cows (n = 1336). Monthly samples were obtained from 38 heifers from birth to one year of age (n = 284 samples). One-half of this research population had been subjected to long-term selection for increased milk yield; high and low forage management groups were overlaid on both the selected and control groups. No TL differences due to selection group or forage management group were noted [75]. Seeker et al. [75] reported a decrease in average TL in leukocytes corresponding to increased age in cows and in heifer calves, with the highest rate of deterioration occurring early in life. Lameness of cows detrimentally influenced TL in their calves [98], as well as heat stress appeared to correspond with lower TL of their progeny [97]. Calves with heavy birth weights had shorter telomeres [75]. TL at birth and one year of age was not associated with cow survival. Telomere attrition over an individual’s life, and especially in early life, was more associated with overall health and culling than average or early life TL [96,97], and the rate of telomere attrition is reported to have a correlation with the life span of a species [115]. However, the permanent environmental variance (of phenotype; traditional quantitative genetics utilizes pedigree relationships to statistically separate this component representing the non-genetic consistency across repeated phenotypes) did not differ from 0 [75]. Seeker et al. [95] reported a quadratic relationship between the additive genetic component (additive contribution of alleles across loci as a contributor to a quantitative phenotype) of TLs with age. Seeker et al. [96] reported 43% of cows that had increased relative TL with age. The average productive lifespan in the cows from their work was just over 4 years; this length, although reasonable in advanced dairy production systems, is biologically very low and may not have permitted a complete assessment of the relationship between TL deterioration and productive longevity. Estimates of heritability for TL were moderate to high (0.36 at birth to 0.46 at first lactation; n = 702 cows; [98]), indicating the additive genetic variance component makes up a moderate proportion of the variance of TL observed. No TL maternal variances differed from 0 [98]. This suggests that, in this population, the genetic factors inherited from the dam do not contribute to the variability of TL. Ilska-Warner et al. [98] reported associations of TL with SNP on BTA 6 and 8. 

The Agerolese dairy breed of Italian origin has a reputation for longevity (approximately 13 years average productive life span) relative to Holstein, which is approximately 4.5 years in the United States. Iannuzzi et al. [99] confirmed a longer average TL in DNA from blood and milk samples of Agerolese cows than in Holstein cows, a negative relationship of TL and cow age in both breeds, and a positive relationship of TL measured in blood samples with that measured in milk samples. 

Tissue differences in TL were also noted in Holstein cows [100,101]. Laubenthal et al. [100] documented a greater TL in liver and subcutaneous adipose samples compared to blood or mammary cell samples. TL was greater in blood and mammary cell samples during early lactation as compared to late lactation. Cows with a greater quantity of telomere products (indicating longer telomeres) in early lactation had greater telomere loss across lactation than other cows. Häussler et al. [101] reported that TL was augmented in cells of the peripheral circulation after lipopolysaccharide administration but was not different in DNA from the liver. Laubenthal et al. [100] concluded that relationships between the quantities in different tissues supported the surrogate use of measures in peripheral blood cells.

The TL of DNA in the semen of Holstein bulls was suggested as a marker for bull semen quality [102,103]. Bulls with “suitable” semen quality had higher volume and motility, increased viability, and a lower incidence of anomalies than bulls in the “unsuitable” category. TL was higher in Holstein bulls with “suitable” semen quality than bulls with “unsuitable” quality. Iannuzzi et al. [102] reported positive correlations of TL with sperm motility and viability and a negative correlation with the frequency of anomalies.

The prenatal environment can have significant biological implications on the fetus that can be observed after birth. Common environmental factors often monitored in dairy production are temperature and humidity. In Holstein–Frisian cows, there was a negative correlation between the median temperature humidity index during the third trimester of gestation and offspring TL [43]. Every one-unit increase of the median third-trimester temperature humidity index led to a 0.35% decrease in TL in the calf. Cow age also influenced TL of their offspring; a negative correlation was found between cow age and TL of the calf.

### 5.2. Beef Cattle

The median TL in DNA from multiple tissues and its variability in Chianina cattle were lower than that of cattle in the Maremmana breed [11]. Although heterosis was not specifically mentioned, the authors noted that the history of the two breeds with respect to straightbreeding or crossbreeding was quite different. The Maremmana breed experienced a population decline across time, and the cattle of that breed in their data were probably the result of crossing with other breeds in their most recent generations; that is, they may have been, in fact, crossbreeds to some extent. This could suggest a heterotic effect on TL. Heterosis is reasonable, as inbreeding has been associated with shorter telomeres in *Mus* and *Peromyscus* species [116,117]. Tilesi et al. [11] reported that the TL of DNA from liver was greater than that from DNA from the lungs and the spleen.

O’Daniel and colleagues [104] investigated the effect of parities, duration of labor, and raising a calf on TL in Brahman beef cattle. Telomeres of cows on their first parity (initial parturtion) were significantly longer than those of age-matched cows on their second parity. TL similarly was less (~4%) in multiparous versus nulliparous women [118]. It was hypothesized that the differences in TL between the 1st parity and 2nd parity cows may be due to the psychological and physiological stress of raising a calf. However, there was no difference in TL before and after calving, and an unexpected positive correlation between labor duration and TL were observed [104]. The hormonal changes and other physiological changes that occur during labor and shortly after could interact to result in no net change in TL. These observations, coupled with the recent report from Panelli et al. [119] that a Cesarean-delivery was associated with reduced leukocyte TL in women, provide additional impetus to study telomere dynamics within dams and their progeny.

Bovine leukemia virus (BLV) can have a severe impact on production and cow longevity and shares several similarities with blood cancers in humans. Telomerase activity in Polish Black cattle was significantly higher in cattle infected with BLV relative to those not infected. Telomerase activity is needed for the maturation of most cancerous tumors. Despite having increased expression of telomerase, telomeres of infected cattle were significantly shorter than those of non-infected cattle [120]. The occurrence of shortened telomeres with high telomerase activity has a strong correlation with disease severity in many blood cancer types. Measuring TL and telomerase activity in affected individuals, cattle, or humans, could be a useful diagnostic mechanism. However, additional knowledge is needed regarding interactions of hormones and cytokines that may affect telomere dynamics by direct modulation of telomerase expression or activity. For example, estradiol stimulates the *telomerase reverse transcriptase* (*TERT*) gene [121], whereas elevated cortisol secretion has been associated with a diminution of TL [122].

A genome-wide association study utilized genotypes from 17 different cattle breeds, including Hereford and Charolais, and estimated leukocyte TL identified 63 statistically significant single nucleotide polymorphisms. The best (lowest *p*-value) signal was located within the *protein tyrosine phosphatase receptor type D* (*PTPRD*). This is the same gene that contained the most significant SNP associated with TL at first lactation in Ilska-Warner et al. [98]. These results suggest that *PTPRD* may be a contributor to the variation in the TL of leukocytes in cattle [123].

### 5.3. Pigs

Research about pig telomeres can be traced back to 2004, when Fradiani et al. [124] reported the medial region of pigs’ telomeres and confirmed the actual sequence as *TTAGGG*, which is similar to the commonly observed telomeric sequence in humans as reported by Brown et al. in 1990 [125]. In general, pig TLs are longer than what humans have, but shorter than those of house mice. TL in porcine cells is gradually gaining more interest from researchers around the world since pigs are proven to be useful as large animal models for studies related to human diseases [126]. Pigs are widely used as models for preclinical studies, because pigs and humans have similar body size, anatomy, diet, and physiology [127,128]. The introduction of genetic engineering to pigs allows scientists to learn more about various diseases like Alzheimer’s disease, cystic fibrosis, Duchenne muscular dystrophy, etc. [129] and to develop pigs for xenotransplantation purposes [130].

Pig telomere dynamics have been utilized to characterize molecular aspects of aging in conjunction with DNA methylation [105,131,132]. The level of DNA methylation is commonly used as an indicator of the aging process in mammals, and one of the examples is the aging of skeletal muscle, which undergoes changes that begin at approximately the mid-point of the human lifespan and become more susceptible to disease [131]. In pigs, the DNA methylation pattern and gene features possess high similarity when compared with that of humans [132]. For putting together the exact changes in DNA methylation at the mid-point of pig lifespan, Jin et al. [105] evaluated Chinese Jinhua pigs at 6 months and 7 years of age to assess the relationship and underlying changes of DNA methylation in skeletal muscles that could be related to the aging process. They measured DNA methylation levels from six types of tissues, including longissimus dorsi muscle, and used quantitative real-time PCR to measure TL. Pigs in the 7-year-old group had a shorter TL in general compared to pigs at 6 months of age [71], which may be explained by hypomethylation in the sub-telomeric regions of the longissimus dorsi muscle, which could be related to muscular atrophy that’s typical for aging muscle [131], from the former group. 

Telomeric regions may also provide useful information for the mechanism of gene expression regulation in porcine gametes. Sperm TL has been suggested as an infertility biomarker for humans [133,134], but it has not been actively studied in pigs until recently. Research has confirmed that telomeres from spermatozoa tend to be longer than those from somatic cells in both humans and pigs [124,135]. Sperm TL in humans is related to important reproductive criteria like spermatogenesis, sperm quality, and fertility [133]. Although the length of telomeres in male germ cells has been shown to be gradually increasing during spermatogenesis, the level of telomerase activity is decreasing and being restricted [136]. There was no significant difference in sperm quality between Pietrain boars with telomeres of different lengths [106]. There were no observed correlations between sperm TL and quality parameters; however, sperm TL does have a correlation with morulae percentage in the early stage of fertilized oocytes, indicating its potential as a biomarker for embryo development in pigs [137]. Telomere size and structure changed through folliculogenesis in pigs, progressively growing and condensing from preantral to antral follicles. This growth was accompanied by enzymatic activity of the telomerase catalytic subunit within the nucleus. Mature oocytes exhibited significantly longer and more stable telomeres than preantral and antral follicles. Telomerase catalytic subunit expression was observed in the ooplasm rather than the nucleus of mature oocytes. Further understanding of telomere elongation and enzymatic activity within developing follicles is important for determining the contribution of proper telomere programing to oocyte quality [73]. 

### 5.4. Sheep

There are fewer reports of TL and related topics in sheep and goats—whether domesticated or wild breeds—than in cattle or pigs. The TL of wild Soay sheep (Hirta Bay, St. Kilda Islands, Scotland, UK) was extensively examined [107,108,109]. Over 20 years of weight records and individual identification of newborn lambs in this wild population each year supported these efforts [138]. Watson et al. [107] reported a sex difference (females greater than males) in TL in the DNA from leukocytes. However, this difference was not detected in sheep younger than 3 years of age. This was consistent with results in humans and mice: mature females tend to have longer telomeres compared to males in the same species [139]. The ambiguous pattern of differences is not clarified until the mid-point of their lifespan; this is consistent with reports from human and mice studies and may possibly be explained by different rates of telomere attrition through development [140]. Froy et al. [108] investigated the source of the association between TL and mortality in Soay sheep. This may be due to genetic effects [141] or environmental effects in early life stages [142]. Froy et al. [108] reported an association, but not causation, between average TL and an individual’s lifespan; they attributed responsibility to the genetic effect because their estimate of the permanent environment effect was practically zero. Additionally, TL across this period was heritable [143] and under the influence of directional natural selection, which implies that the TL is capable of evolving under natural conditions [108]. A decrease in TL is suspected to be related to a heavy reproductive investment in females, meaning that females giving birth to any offspring may have shorter TL on average compared to those who did not breed [144]. However, females with a heavy reproductive investment like gestation and lactation had longer TL than females with investment in gestation only (documented when the offspring were lost after birth due to various reasons [109]).

### 5.5. Horses

The average TL in younger horses (10 years of age or less) was 16.9 kb, and in aged subjects (25 to 30 years of age, much longer than other livestock species), the average was 13.6 kb [12]. As in other livestock species, strong inverse correlations between TL and age have been observed. In Thoroughbred horses, the average annual shortening of telomeres from peripheral blood mononuclear cells was 134 bp per year [111]. There was substantial inter-animal variability in TL of younger animals [110]. The variability in growth rates in younger animals, health status at the time of collection, and genetics could contribute to the high variability in leukocyte TL. There were no statistically significant correlations between TL and performance-based (competitive racing) history records (race winnings, the number of races entered, or minimum or maximum distances of entered races). There were no associations between TL and other economically relevant traits in the equine industry, such as sex or coat color [110]. 

Telomere shortening is thought to contribute to lymphocyte immunosenescence, leading to decreased immune function as an individual ages. Various parameters of immune function and their relationship with TL were investigated in a group of mixed-breed horses. A positive correlation was observed between in vitro proliferation of peripheral blood mononuclear cells and TL, as well as total serum Immunoglobulin G. There was a negative correlation between TL and the percentage of lymphocytes that produce interferon-gamma and monocytes that produce tumor necrosis factor-alpha [112]. However, in a population of aged horses (20 years of age and older), there were no statistically significant correlations between TL and immunological/inflammatory responses. If horses exhibiting elevated inflammation and shorter telomeres are less likely to live longer, this could potentially lead to an underestimation of the association between inflammation and telomere length in aging horses. Further investigation is needed to identify the impact that telomere shortening has on immune response in conjunction with other factors such as previous environmental exposures and host genetics [145]. 

It is thought that TL could be used as a possible indicator of health and welfare in horses [113]. It was hypothesized that horses that exhibit a high occurrence of abnormal oral behaviors (chewing, excessive licking, and bedding consumption) would have differing TLs than those with a low occurrence. However, no differences in the TL of horses were detected between the groups. Since diet and lifestyle influence TL, the effect of diet change on TL in horses with abnormal oral behaviors was also investigated. Diets consisting of higher forage content are thought to be more beneficial to horse welfare and have reduced abnormal oral behaviors relative to a high-concentrate low-forage diet. When a high-concentrate diet was replaced with a high-forage diet, there were no significant differences between the TL pre- and post-diet change. There was no difference in TL between horses exhibiting a high occurrence of abnormal oral behavior post-diet change length and those of horses with a low occurrence of abnormal oral behaviors [114].

## 6. Future Considerations

There are several considerations to prioritize as the study of telomere dynamics in livestock goes forward. As discussed in the preceding sections of this review, advancement in the quest to selectively manipulate telomere dynamics of livestock in order to improve their well-being and lifetime productivity is constrained by technical considerations and comprehension of intrinsic and extrinsic factors that modulate TL. Recently, Vaiserman and Krasnienkov [4] emphasized that the application of TL as the predominant marker in the study of human aging is limited by an incomplete recognition of the impact of developmental programming, cell lineage, and environmental exposures on telomere dynamics. Relative to the more extensive knowledge base established for humans, laboratory rodents, and some avians, fundamental details of telomere dynamics in livestock species are limited. What is known about TL for livestock species is primarily derived from the study of dairy cattle. Due to the tractability of dairy cows and the economic necessity that they have a productive lifespan (i.e., deliver a calf annually and lactate at an above-average yield of high-quality milk), the dairy cow has been the predominant animal model to investigate factors that influence TL in livestock. Human health considerations constitute the primary impetus to develop research tools to advance the understanding of telomere dynamics. Bhala and Savage [146] reviewed the comparative advantages of TL measurement options on the current menu of methods available for human clinical genetic counseling. Regarding the assessment of telomere dynamics in dairy cattle and the meat-producing species, there must be a concerted effort to minimize the lag time in the application of the technology reviewed [146]. Moreover, properly designed and resourced longitudinal studies of livestock species that apply appropriate methodology are needed to assess telomere dynamics beyond quantification of the average TL. Through such studies, it may be possible to ascertain with greater clarity the mechanisms whereby telomere dynamics may modulate livestock health, fertility, and/or productivity. 

In general, the potential relationship of TL maintenance with the biogenesis and efficiency of mitochondria developing and mature somatic and germ cells merits deeper investigation in view of the inhibitory effect of stress-related hormones on mitochondrial efficiency [40,147]. The concept that TL is affected by environmental stimuli in early postnatal life has been extended to include not only the prenatal developmental phase but also the periconception phase [32]. Demanelis et al. [45] reported, as part of the Genotype-Tissue Expression (GTEx) project [148] that: (1) relative TL of tissues were positively correlated (low to moderate correlation coefficients) for adult tissues that were derived from the same embryonic germ layer (i.e., ectoderm, endoderm, mesoderm); and (2) TL of peripheral white blood cells could be considered as a surrogate for TL of specific cell types (correlation coefficients ranged from 0.15 to 0.37). This range of correlation coefficients between leukocyte TL and various solid tissue sources of rats was recently reported by Semeraro et al. [149]. Although there are a few reports regarding the TL of liver, mammary, and sperm cells from dairy cattle [100,101,102], the questions of telomere dynamics within specific tissue types (epithelial, neural, connective, and muscular) and concordance of TL between peripheral blood cell types and specific somatic tissues of livestock species remain open.

## 7. Summary

As is frequently the situation in livestock species, the discovery, and clarification of the activities of telomere sequences somewhat follow that found in humans. Early research on clone longevity and survival in several species instigated the early molecular knowledge of telomeres in livestock. There has been no consensus result regarding clone telomere attrition or gain across species or tissues. Across-generation TL correspondence, as detailed in humans, may be an opportunity for improvement strategies in livestock. Stress appears to negatively impact telomere maintenance in livestock species. Correspondence of longevity with telomere attrition in livestock is of great interest as successful genetic improvement of longevity traits has been limited to date. The definition of longevity traits is not consistent across livestock species (longevity in horses is much longer than that of other species), nor even within species (e.g., dairy vs. beef production). Research so far has mostly focused on dairy cattle; future studies may consider investigating under-characterized species to better understand telomere dynamics in livestock. Telomere dynamics are particularly under-characterized in beef cattle, domesticated sheep, and goats. Elucidation of livestock telomere dynamics within species and production systems appears to be scientifically worthwhile.

## Figures and Tables

**Table 1 biology-12-01389-t001:** Telomere length investigations in cloned livestock.

Species	Donor Cell	Method	Clone Relative to Control	Source
Bovine	Cumulus and fibroblast	Southern blot: terminal restriction fragment	Same length	[61]
	Somatic	Southern blot: terminal restriction fragment	Longer	[62]
	Fibroblast	Southern blot analysis	Same length	[63]
	Muscle		Same length	
	Oviductal epithelial		Shorter	
	Mammary epithelial		Shorter	
Porcine	Fetal fibroblast	Terminal restriction fragment	Same length	[64]
	Fetal fibroblast	Terminal restriction fragment	Longer	[65]
	Ear skin fibroblast	qPCR	Longer	[66]
Ovine	Mammary epithelial	Southern blot: terminal restriction fragment	Shorter	[67]
	Embryonic cells		Shorter	
	Fetal fibroblast	Southern blot: terminal restriction fragment	Same length	[68]
Caprine	Fetal fibroblast	Southern blot: terminal restriction fragment	Shorter	[69]
	Ear skin adult fibroblast	Southern blot: terminal restriction fragment	Shorter	[70]

**Table 2 biology-12-01389-t002:** Livestock telomere length (TL) investigations.

Species	Assay	Major Findings	Sources
*Bos taurus* dairy	Monochrome multiplex quantitative PCR (qPCR)	Shorter TL in older cows and cows with shorter productive life Absence of breed differences. TL associated with improved performance/health. TL not associated with longevity or fitness. Heritability estimates: 0.12 to 0.2.	[92,93,94]
	qPCR calibrated for DNA extraction method	TL and age in general are inversely related, but not always. Stress induces telomere attrition. Early life: more rapid TL attrition. Rate of attrition may be more important than absolute TL for association with life productivity traits. Heritability estimates: 0.36 to 0.46.	[75,95,96,97]
	qPCR	Breed differences detected. Negative relationship of TL with age. Correspondence of TL measured in blood and milk.	[98]
	qPCR	Tissue differences in TL detected. Late lactation samples had lower TL than early lactation samples.	[99]
	qPCR	Tissue differences in TL detected.	[100,101]
	qPCR	Sperm DNA: TL greater in good quality semen samples than in low quality.	[102,103]
*Bos taurus* beef	Densitometry	Breed differences detected. Heterosis may be influential. Tissue differences detected.	[11]
*Bos indicus* beef	qPCR	Negative relationship of TL with parity. No difference in TL before and after parturition.	[104]
*Sus scrofus*	qPCR	Age effect (TL shorter in aged pigs) may be related to methylation of DNA.	[105]
	qFISH	No TL differences detected in spermatozoa DNA. Spermatozoa TL positively associated with embryonic development of oocytes.	[106]
*Ovis aries*	qPCR	Shorter TL in older wild (Soay) sheep. Stress of reproductive investment associated with shorter TL.	[107,108,109]
*Equus caballus*	Telomere restriction fragment (TRF)	Shorter TL in older horses.	[12]
	qPCR	Shorter TL in older horses. No association of TL with racing performance. No association of TL with sex or coat color.	[110,111]
	Fluorescence-based in situ hybridization	Limited evidence of association of TL with immune response.	[112]
	qPCR	No detected TL between horses with abnormal oral behaviors nor diet.	[113,114]

## Data Availability

Not applicable.

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
