# Peer review of "Telomere Dynamics in Livestock"

_biology, 2023, doi:10.3390/biology12111389_

Round 1

Reviewer 1 Report

Comments and Suggestions for Authors

My comments are summarized in the attached review. 

Author Response

See the accompanying file for each suggestion.

Reviewer 2 Report

Comments and Suggestions for Authors

In manuscript, the authors reviewed the progress of the studies on telomere dynamics in livestock. It is informative and helpful for the readers to understanding the status in this field.

I have following suggestions for the authors.

1.     In many studies, it is indicated that telomere length is associated with longevity. Are there any specific studies on the mechanism of the association?  Similarly, for other potential bio-functions related to the telomere length, it will be ideal to provide possible explanations.

2.     In the last paragraph, it is suggested to provide the prospect of the further studies in this field, such as the key issues remained to make clear. It will be helpful to the readers.\

Comments on the Quality of English Language

   The authors defined the telomere length as “TL”, and only used the abbreviated form in Table 2.  The abbreviation should be used in the whole main text.

Author Response

See accompanying file for responses to comments.

Reviewer 3 Report

Comments and Suggestions for Authors

In the present review manuscript, the authors highlighted the importance of telomere dynamics in livestock species, including the aging process and SCNT outcomes. The authors comprehensively described current knowledge on telomere investigations in livestock species, including an in-depth description of methods used to determine the telomere length and telomer research in livestock. The special section of telomere studies in cloned animals will be very useful to researchers working on SCNT applications. I suggest having one section on telomer and its association with production and health traits in livestock.  Overall, this review will be very useful for researchers who are working in telomere biology in livestock.  

Author Response

See the accompanying file for responses.
